# Geometric optimisation on positive definite matrices with application to elliptically contoured distributions

**Suvrit Sra**
Max Planck Institute for Intelligent Systems
Tübingen, Germany

**Reshad Hosseini**
School of ECE, College of Engineering
University of Tehran, Tehran, Iran

## Abstract

Hermitian positive definite (hpd) matrices recur throughout machine learning, statistics, and optimisation. This paper develops (conic) *geometric optimisation* on the cone of hpd matrices, which allows us to globally optimise a large class of nonconvex functions of hpd matrices. Specifically, we first use the Riemannian manifold structure of the hpd cone for studying functions that are nonconvex in the Euclidean sense but are geodesically convex (g-convex), hence globally optimisable. We then go beyond g-convexity, and exploit the conic geometry of hpd matrices to identify another class of functions that remain amenable to global optimisation without requiring g-convexity. We present key results that help recognise g-convexity and also the additional structure alluded to above. We illustrate our ideas by applying them to likelihood maximisation for a broad family of elliptically contoured distributions: for this maximisation, we derive novel, parameter free fixed-point algorithms. To our knowledge, ours are the most general results on geometric optimisation of hpd matrices known so far. Experiments show that advantages of using our fixed-point algorithms.

## 1 Introduction

The geometry of Hermitian positive definite (hpd) matrices is remarkably rich and forms a foundational pillar of modern convex optimisation [21] and of the rapidly evolving area of convex algebraic geometry [4]. The geometry exhibited by hpd matrices, however, goes beyond what is typically exploited in these two areas. In particular, hpd matrices form a convex cone which is also a differentiable Riemannian manifold that is also a CAT(0) space (i.e., a metric space of nonpositive curvature [7]). This rich structure enables "geometric optimisation" with hpd matrices, which allows solving many problems that are nonconvex in the Euclidean sense but convex in the manifold sense (see §2 or [29]), or have enough metric structure (see §3) to permit efficient optimisation.

This paper develops (conic) *geometric optimisation*[1] (GO) for hpd matrices. We present key results that help recognise geodesic convexity (g-convexity); we also present sufficient conditions that put a class of even non g-convex functions within the grasp of GO. To our knowledge, ours are the most general results on geometric optimisation with hpd matrices known so far.

**Motivation for GO.** We begin by noting that the widely studied class of *geometric programs* is ultimately nothing but the 1D version of GO on hpd matrices. Given that geometric programming has enjoyed great success in numerous applications—see e.g., the survey of Boyd et al. [6]—we hope GO also gains broad applicability. For this paper, GO arises naturally while performing maximum likelihood parameter estimation for a rich class of *elliptically contoured distributions*

(ECDs) [8, 13, 20]. Perhaps the best known GO problem is the task of computing the Karcher / Fréchet-mean of hpd matrices: a topic that has attracted great attention within matrix theory [2, 3, 27], computer vision [10], radar imaging [22; Part II], and medical imaging [11, 31]—we refer the reader to the recent book [22] for additional applications, references, and details. Another GO problem arises as a subroutine in nearest neighbour search over hpd matrices [12]. Several other areas involve GO problems: statistics (covariance shrinkage) [9], nonlinear matrix equations [17], Markov decision processes and the wider encompassing area of nonlinear Perron-Frobenius theory [18].

**Motivating application.** We use ECDs as a platform for illustrating our ideas for two reasons: (i) ECDs are important in a variety of settings (see the recent survey [23]); and (ii) they offer an instructive setup for presenting key ideas from the world of geometric optimisation.

Let us therefore begin by recalling some basics. An ECD with density on $\mathbb{R}^d$ takes the form [2]

$$\forall\, x \in \mathbb{R}^d, \qquad \mathscr{E}_\varphi(x; S) \propto \det(S)^{-1/2} \varphi(x^T S^{-1} x), \tag{1}$$

where $S \in \mathbb{P}_d$ (i.e., the set of $d \times d$ symmetric positive definite matrices) is the scatter matrix while $\varphi : \mathbb{R} \to \mathbb{R}_{++}$ is positive *density generating function* (dgf). If ECDs have finite covariance matrix, then the scatter matrix is proportional to the covariance matrix [8].

**Example 1.** With $\varphi(t) = e^{-\frac{t}{2}}$, density (1) reduces to the multivariate normal density. For the choice

$$\varphi(t) = t^{\alpha - d/2} \exp\left(-(t/b)^\beta\right), \tag{2}$$

where $\alpha$, $b$ and $\beta$ are fixed positive numbers, density (1) yields the rich class called *Kotz-type distributions* that are known to have powerful modelling abilities [15; §3.2]; they include as special cases multivariate power exponentials, elliptical gamma, multivariate W-distributions, for instance.

**MLE.** Let $(x_1, \ldots, x_n)$ be i.i.d. samples from an ECD $\mathscr{E}_\varphi(S)$. Up to constants, the log-likelihood is

$$\mathcal{L}(S) = -\tfrac{1}{2} n \log \det S + \sum\nolimits_{i=1}^{n} \log \varphi(x_i^T S^{-1} x_i). \tag{3}$$

Equivalently, we may consider the minimisation problem

$$\min_{S \succ 0} \quad \Phi(S) := \tfrac{1}{2} n \log \det(S) - \sum\nolimits_i \log \varphi(x_i^T S^{-1} x_i). \tag{4}$$

Problem (4) is in general difficult as $\Phi$ may be nonconvex and may have multiple local minima. Since statistical estimation theory relies on having access to global optima, it is important to be able to solve (4) to global optimality. These difficulties notwithstanding, using GO ideas, we identify a rich class of ECDs for which we can indeed solve (4) optimally. Some examples already exist in the literature [16, 23, 30]; this paper develops techniques that are strictly more general and subsume previous examples, while advancing the broader idea of geometric optimisation.

We illustrate our ideas by studying the following two main classes of dgfs in (1):

(i) **Geodesically Convex (GC):** This class contains functions for which the negative log-likelihood $\Phi(S)$ is g-convex, i.e., convex along geodesics in the manifold of hpd matrices. Some members of this class have been previously studied (though sometimes without recognising or directly exploiting the g-convexity);

(ii) **Log-Nonexpansive (LN):** This is a new class that we introduce in this paper. It exploits the "non-positive curvature" property of the manifold of hpd matrices.

There is a third important class: LC, the class of log-convex dgfs $\varphi$. Though, since (4) deals with $-\log \varphi$, the optimisation problem is still nonconvex. We describe class LC only in [28] primarily due to paucity of space and also because the first two classes contain our most novel results. These classes of dgfs are neither mutually disjoint nor proper subsets of each other. Each captures unique analytic or geometric structure crucial to efficient optimisation. Class GC characterises the "hidden" convexity found in several instances of (4), while LN is a novel class of models that might not have this hidden convexity, but nevertheless admit global optimisation.

**Contributions.** The key contributions of this paper are the following:

– New results that characterise and help recognise g-convexity (Thm. 1, Cor. 2, Cor. 3, Thm. 4). Though initially motivated by ECDs, our matrix-theoretic proofs are more generally applicable and should be of wider interest. All technical proofs, and several additional results that help recognise g-convexity are in the longer version of this paper [28].

– New fixed-point theory for solving GO problems, including some that might even lack g-convexity. Here too, our results go beyond ECDs—in fact, they broaden the class of problems that admit fixed-point algorithms in the metric space $(\mathbb{P}_d, \delta_T)$—Thms. 11 and 14 are the key results here.

Our results on geodesic convexity subsume the more specialised results reported recently in [29]. We believe our matrix-theoretic proofs, though requiring slightly more advanced machinery, are ultimately simpler and more widely applicable. Our fixed-point theory offers a unified framework that not only captures the well-known M-estimators of [16], but applies to a larger class of problems than possible using previous methods. Our experimental illustrate computational benefits of one of resulting algorithms.

## 2  Geometric optimisation with geodesic convexity: class GC

Geodesic convexity (g-convexity) is a classical concept in mathematics and is used extensively in the study of Hadamard manifolds and metric spaces of nonpositive curvature [7, 24] (i.e., spaces whose distance function is g-convex). This concept has been previously studied in nonlinear optimisation [25], but its full importance and applicability in statistical applications and optimisation is only recently emerging [29, 30].

We begin our presentation by recalling some definitions—please see [7, 24] for extensive details.

**Definition 2** (gc set). Let $\mathcal{M}$ denote a $d$-dimensional connected $C^2$ Riemannian manifold. A set $\mathcal{X} \subset \mathcal{M}$, where is called *geodesically convex* if any two points of $\mathcal{X}$ are joined by a geodesic lying in $\mathcal{X}$. That is, if $x, y \in \mathcal{X}$, then there exists a path $\gamma : [0,1] \to \mathcal{X}$ such that $\gamma(0) = x$ and $\gamma(1) = y$.

**Definition 3** (gc function). Let $\mathcal{X} \subset \mathcal{M}$ be a gc set. A function $\phi : \mathcal{X} \to \mathbb{R}$ is *geodesically convex*, if for any $x, y \in \mathcal{X}$ and a unit speed geodesic $\gamma : [0,1] \to \mathcal{X}$ with $\gamma(0) = x$ and $\gamma(1) = y$, we have

$$\phi(\gamma(t)) \leq (1-t)\phi(\gamma(0)) + t\phi(\gamma(1)) = (1-t)\phi(x) + t\phi(y). \tag{5}$$

The power of gc functions in the context of solving (4) comes into play because the set $\mathbb{P}_d$ (the convex cone of positive definite matrices) is also a differentiable Riemannian manifold where geodesics between points can be computed efficiently. Indeed, the tangent space to $\mathbb{P}_d$ at any point can be identified with the set of Hermitian matrices, and the inner product on this space leads to a Riemannian metric on $\mathbb{P}_d$. At any point $A \in \mathbb{P}_d$, this metric is given by the differential form $ds = \|A^{-1/2}dAA^{-1/2}\|_F$; also, between $A, B \in \mathbb{P}_d$ there is a unique geodesic [1; Thm. 6.1.6]

$$A\#_t B := \gamma(t) = A^{1/2}(A^{-1/2}BA^{-1/2})^t A^{1/2}, \quad t \in [0,1]. \tag{6}$$

The midpoint of this path, namely $A\#_{1/2}B$ is called the *matrix geometric mean*, which is an object of great interest in numerous areas [1–3, 10, 22]. As per convention, we denote it simply by $A\#B$.

**Example 4.** Let $z \in \mathbb{C}^d$ be any vector. The function $\phi(X) := z^*X^{-1}z$ is gc.
*Proof.* Since $\phi$ is continuous, it suffices to verify midpoint convexity: $\phi(X\#Y) \leq \frac{1}{2}\phi(X) + \frac{1}{2}\phi(Y)$, for $X, Y \in \mathbb{P}_d$. Since $(X\#Y)^{-1} = X^{-1}\#Y^{-1}$ and $X^{-1}\#Y^{-1} \preceq \frac{X^{-1}+Y^{-1}}{2}$ ([1; 4.16]), it follows that $\phi(X\#Y) = z^*(X\#Y)^{-1}z \leq \frac{1}{2}(z^*X^{-1}z + z^*Y^{-1}z) = \frac{1}{2}(\phi(X) + \phi(Y))$.

We are ready to state our first main theorem, which vastly generalises the above example and provides a foundational tool for recognising and constructing gc functions.

**Theorem 1.** *Let $\Pi : \mathbb{P}_d \to \mathbb{P}_k$ be a strictly positive linear map. Let $A, B \in \mathbb{P}_d$ we have*

$$\Pi(A\#_t B) \preceq \Pi(A)\#_t\Pi(B), \qquad t \in [0,1]. \tag{7}$$

*Proof.* Although positive linear maps are well-studied objects (see e.g., [1; Ch. 4]), we did not find an explicit proof of (7) in the literature, so we provide a proof in the longer version [28].  □

A useful corollary of Thm. 1 is the following (notice this corollary subsumes Example 4).

**Corollary 2.** *For positive definite matrices $A, B \in \mathbb{P}_d$ and matrices $0 \neq X \in \mathbb{C}^{d \times k}$ we have*

$$\mathrm{tr}\, X^*(A\#_t B)X \leq [\mathrm{tr}\, X^*AX]^{1-t}[\mathrm{tr}\, X^*BX]^t, \qquad t \in (0,1). \tag{8}$$

*Proof.* Use the map $A \mapsto \operatorname{tr} X^* A X$ in Thm. 1. □

**Note:** Cor. 2 actually constructs a log-g-convex function, from which g-convexity is immediate.

A notable corollary to Thm. 1 that subsumes a nontrivial result [14; Lem. 3.2] is mentioned below.

**Corollary 3.** *Let $X_i \in \mathbb{C}^{d \times k}$ with $k \leq d$ such that $\operatorname{rank}([X_i]_{i=1}^m) = k$. Then the function $\phi(S) := \log \det(\sum_i X_i^* S X_i)$ is gc on $\mathbb{P}_d$.*

*Proof.* By our assumption on the $X_i$, the map $\Pi = S \mapsto \sum_i X_i^* S X_i$ is strictly positive. Thus, from Thm 1 it follows that $\Pi(S \# R) = \sum_i X_i^* (S \# R) X_i \preceq \Pi(S) \# \Pi(R)$. Since $\log \det$ is monotonic, and determinant is multiplicative, the previous inequality yields

$$\phi(S \# R) = \log \det \Pi(S \# R) \leq \log \det(\Pi(S)) + \log \det(\Pi(R)) = \tfrac{1}{2}\phi(S) + \tfrac{1}{2}\phi(R). \qquad \square$$

We are now ready to state our second main theorem.

**Theorem 4.** *Let $h : \mathbb{P}_k \to \mathbb{R}$ be gc function that is nondecreasing in Löwner order. Let $r \in \{\pm 1\}$, and let $\Pi : \mathbb{P}_d \to \mathbb{P}_k$ be a strictly positive linear map. Then, $\phi(S) = h(\Pi(S^r)) \pm \log \det(S)$ is gc.*

*Proof.* Since $\phi$ is continuous, it suffices to prove midpoint geodesic convexity. Since $r \in \{\pm 1\}$, $(S \# R)^r = S^r \# R^r$; thus, from Thm. 1 and since $h$ is matrix nondecreasing, it follows that

$$h(\Pi(S \# R)^r) = h(\Pi(S^r \# R^r)) \leq h(\Pi(S^r) \# \Pi(R^r)). \tag{9}$$

Since $h$ is also gc, inequality (9) further yields

$$h(\Pi(S^r) \# \Pi(R^r)) \leq \tfrac{1}{2} h(\Pi(S^r)) + \tfrac{1}{2} h(\Pi(R^r)). \tag{10}$$

Since $\pm \log \det(S \# R) = \pm \tfrac{1}{2}\big(\log \det(S) + \log \det(R)\big)$, on combining with (10) we obtain

$$\phi(S \# R) \leq \tfrac{1}{2}\phi(S) + \tfrac{1}{2}\phi(R),$$

as desired. Notice also that if $h$ is strictly gc, then $\phi(S)$ is also strictly gc. □

Finally, we state a corollary of Thm. 4 helpful towards recognising geodesic convexity of ECDs. We mention here that a result equivalent to Corr. 5 was recently also discovered in [30]. Thm. 4 is more general and uses a completely different argument founded on the matrix-theoretic results; our techniques may also be of wider independent interest.

**Corollary 5.** *Let $h : \mathbb{R}_{++} \to \mathbb{R}$ be nondecreasing and gc (i.e., $h(x^{1-\lambda} y^\lambda) \leq (1-\lambda)h(x) + \lambda h(y)$). Then, for $r \in \{\pm 1\}$, $\phi : \mathbb{P}_d \to \mathbb{R} : S \mapsto \sum_i h(x_i^T S^r x_i) \pm \log \det(S)$ is gc.*

## 2.1 Application to ECDs in class GC

We begin with a straightforward corollary of the above discussion.

**Corollary 6.** *For the following distributions the negative log-likelihood (4) is gc: (i) Kotz with $\alpha \leq \frac{d}{2}$ (its special cases include Gaussian, multivariate power exponential, multivariate W-distribution with shape parameter smaller than one, elliptical gamma with shape parameter $\nu \leq \frac{d}{2}$); (ii) Multivariate-t; (iii) Multivariate Pearson type II with positive shape parameter; (iv) Elliptical multivariate logistic distribution.* [3]

If the log-likelihood is strictly gc then (4) cannot have multiple solutions. Moreover, for any local optimisation method that computes a solution to (4), geodesic convexity ensures that this solution is globally optimal. Therefore, the key question to answer is: (i) *does* (4) *have a solution?*

Note that answering this question is nontrivial even in special cases [16, 30]. We provide below a fairly general result that helps establish existence.

**Theorem 7.** *If $\Phi(S)$ satisfies the following properties: (i) $-\log\varphi(t)$ is lower semi-continuous (lsc) for $t > 0$, and (ii) $\Phi(S) \to \infty$ as $\|S\| \to \infty$ or $\|S^{-1}\| \to \infty$, then $\Phi(S)$ attains its minimum.*

*Proof.* Consider the metric space $(\mathbb{P}_d, d_R)$, where $d_R$ is the Riemannian distance,

$$d_R(A, B) = \|\log(A^{-1/2}BA^{-1/2})\|_F \qquad A, B \in \mathbb{P}_d. \tag{11}$$

If $\Phi(S) \to \infty$ as $\|S\| \to \infty$ or as $\|S^{-1}\| \to \infty$, then $\Phi(S)$ has bounded lower-level sets in $(\mathbb{P}_d, d_R)$. It is a well-known result in variational analysis that a function that has bounded lower-level sets in a metric space and is lsc, then the function attains its minimum [26]. Since $-\log\varphi(t)$ is lsc and $\log\det(S^{-1})$ is continuous, $\Phi(S)$ is lsc on $(\mathbb{P}_d, d_R)$. Therefore it attains its minimum. $\qquad\square$

A key consequence of Thm. 7 is its ability to show existence of solutions to (4) for a variety of different ECDs. Let us look at an application to Kotz-type distributions below. For these distributions, the function $\Phi(S)$ assumes the form

$$K(S) = \tfrac{n}{2}\log\det(S) + (\tfrac{d}{2} - \alpha)\sum_{i=1}^{n}\log x_i^T S^{-1} x_i + \sum_{i=1}^{n}\left(\tfrac{x_i^T S^{-1} x_i}{b}\right)^{\beta}. \tag{12}$$

Lemma 8 shows that $K(S) \to \infty$ whenever $\|S^{-1}\| \to \infty$ or $\|S\| \to \infty$.

**Lemma 8.** *Let the data $\mathcal{X} = \{x_1, \ldots, x_n\}$ span the whole space and satisfy for $\alpha < \tfrac{d}{2}$ the condition*

$$\frac{|\mathcal{X} \cap L|}{|\mathcal{X}|} < \frac{d_L}{d - 2\alpha}, \tag{13}$$

*where $L$ is an arbitrary subspace with dimension $d_L < d$ and $|\mathcal{X} \cap L|$ is the number of datapoints that lie in the subspace $L$. If $\|S^{-1}\| \to \infty$ or $\|S\| \to \infty$, then $K(S) \to \infty$.*

*Proof.* If $\|S^{-1}\| \to \infty$ and since the data span the whole space, it is possible to find a datum $x_1$ such that $t_1 = x_1^T S^{-1} x_1 \to \infty$. Since

$$\lim_{t \to \infty} c_1 \log(t) + t^{c_2} + c_3 \to \infty$$

for constants $c_1, c_3$ and $c_2 > 0$, it follows that $K(S) \to \infty$ whenever $\|S^{-1}\| \to \infty$.

If $\|S\| \to \infty$ and $\|S^{-1}\|$ is bounded, then the third term in expression of $K(S)$ is bounded. Assume that $d_L$ is the number of eigenvalues of $S$ that go to $\infty$ and $|\mathcal{X} \cap L|$ is the number of data that lie in the subspace span by these eigenvalues. Then in the limit when eigenvalues of $S$ go to $\infty$, $K(S)$ converges to the following limit

$$\lim_{\lambda \to \infty} \tfrac{n}{2}d_L\log\lambda + (\tfrac{d}{2} - \alpha)|\mathcal{X} \cap L|\log\lambda^{-1} + c$$

Apparently if $\tfrac{n}{2}d_L - (\tfrac{d}{2} - \alpha)|\mathcal{X} \cap L| > 0$, then $K(S) \to \infty$ and the proof is complete. $\qquad\square$

It is important to note that overlap condition (13) can be fulfilled easily by assuming that the number of data is larger than their dimensionality and that they are noisy. Using Lemma 8, we can invoke Thm. 7 to immediately state the following result.

**Theorem 9** (Existence Kotz-distr.). *If the data samples satisfy condition* (13)*, then the Kotz negative log-likelihood has a minimiser.*

As previously mentioned, once existence is ensured, one may use any local optimisation method to minimise (4) to obtain the desired mle. This brings us to the next question. What if $\Phi(S)$ is neither convex nor g-convex? The ideas introduced in Sec. 3 below offer a partial one answer.

## 3 Geometric optimisation for class LN

Without convexity or g-convexity, in general at best we might obtain local minima. However, as alluded to previously, the set $\mathbb{P}_d$ of hpd matrices possesses remarkable geometric structure that allows us to extend global optimisation to a rich class beyond just gc functions. To our knowledge, this class of ECDs was beyond the grasp of previous methods [16, 29, 30]. We begin with a key definition.

**Definition 5** (Log-nonexpansive). Let $f : \mathbb{R}_{++} \to (0, \infty)$. We say $f$ is *log-nonexpansive* (LN) on a compact interval $I \subset \mathbb{R}_+$ if there exists a *fixed* constant $0 \leq q \leq 1$ such that

$$|\log f(t) - \log f(s)| \leq q|\log t - \log s|, \quad \forall s, t \in I. \tag{14}$$

If $q < 1$, we say $f$ is *log-contractive*. Finally, if for every $s \neq t$ it holds that

$$|\log f(t) - \log f(s)| < |\log t - \log s|, \quad \forall s, t \quad s \neq t,$$

we say $f$ is *weakly log-contractive* (wlc); an important point to note here is the absence of a fixed $q$.

Next we study existence, uniqueness, and computation of solutions to (4). To that end, momentarily ignore the constraint $S \succ 0$, to see that the first-order necessary optimality condition for (4) is

$$\frac{\partial \Phi(S)}{\partial S} = 0 \quad \Longleftrightarrow \quad \frac{1}{2} n S^{-1} + \sum_{i=1}^{n} \frac{\varphi'(x_i^T S^{-1} x_i)}{\varphi(x_i^T S^{-1} x_i)} S^{-1} x_i x_i^T S^{-1} = 0. \tag{15}$$

Defining $h \equiv -\varphi'/\varphi$, condition (15) may be rewritten more compactly as

$$S = \frac{2}{n} \sum_{i=1}^{n} x_i h(x_i^T S^{-1} x_i) x_i^T = \frac{2}{n} X h(D_S) X^T, \tag{16}$$

where $D_S := \text{Diag}(x_i^T S^{-1} x_i)$, and $X = [x_1, \ldots, x_m]$. If (16) has a positive definite solution, then it is a candidate mle; if it is unique, then it is the desired solution (observe that if we have a Gaussian, then $h(t) \equiv 1/2$, and as expected (16) reduces to the sample covariance matrix).

But how should we solve (16)? This question is in general highly nontrivial to answer because (16) is difficult nonlinear equation in matrix variables. This is the point where the class LN introduced above comes into play. More specifically, we solve (16) via a fixed-point iteration. Introduce therefore the nonlinear map $\mathcal{G} : \mathbb{P}_d \to \mathbb{P}_d$ that maps $S$ to the right hand side of (16); then, starting with a feasible $S_0 \succ 0$, simply perform the iteration

$$S_{k+1} \leftarrow \mathcal{G}(S_k), \quad k = 0, 1, \ldots, \tag{17}$$

which is shown more explicitly as Alg. 1 below.

---

**Algorithm 1** Fixed-point iteration for mle

*Input:* Observations $x_1, \ldots, x_n$; function $h$
*Initialize:* $k \leftarrow 0$; $S_0 \leftarrow I_n$
**while** $\neg$ converged **do**
$\quad S_{k+1} \leftarrow \frac{2}{n} \sum_{i=1}^{n} x_i h(x_i^T S_k^{-1} x_i) x_i^T$
**end while**
**return** $S_k$

---

The most interesting twist to analysing iteration (17) is that the map $\mathcal{G}$ is usually *not* contractive with respect to the Euclidean metric. But the metric geometry of $\mathbb{P}_d$ alluded to previously suggests that it might be better to analyse the iteration using a non-Euclidean metric. Unfortunately, the Riemannnian distance (11) on $\mathbb{P}_d$, while canonical, also turns out to be unsuitable. This impasse is broken by selecting a more suitable "hyperbolic distance" that captures the crucial non-Euclidean geometry of $\mathbb{P}_d$, while still respecting its convex conical structure.

Such a suitable choice is provided by the Thompson metric—an object of great interest in nonlinear matrix equations [17]—which is known to possess geometric properties suitable for analysing convex cones, of which $\mathbb{P}_d$ is a shining example [18]. On $\mathbb{P}_d$, the *Thompson metric* is given by

$$\delta_T(X, Y) := \|\log(Y^{-1/2} X Y^{-1/2})\|, \tag{18}$$

where $\|\cdot\|$ is the usual operator 2-norm, and 'log' is the matrix logarithm. The core properties of (18) that prove useful for analysis fixed point iterations are listed below—for proofs please see [17, 19].

**Proposition 10.** *Unless noted otherwise, all matrices are assumed to be hpd..*

$$\delta_T(X^{-1}, Y^{-1}) = \delta_T(X, Y) \tag{19a}$$

$$\delta_T(B^* X B, B^* Y B) = \delta_T(X, Y), \quad B \in GL_n(\mathbb{C}) \tag{19b}$$

$$\delta_T(X^t, Y^t) \leq |t|\delta_T(X, Y), \quad for \ t \in [-1, 1] \tag{19c}$$

$$\delta_T\left(\sum_i w_i X_i, \sum_i w_i Y_i\right) \leq \max_{1 \leq i \leq m} \delta_T(X_i, Y_i), \quad w_i \geq 0, w \neq 0 \tag{19d}$$

$$\delta_T(X + A, Y + A) \leq \frac{\alpha}{\alpha + \beta} \delta_T(X, Y), \quad A \succeq 0, \tag{19e}$$

*where* $\alpha = \max\{\|X\|, \|Y\|\}$ *and* $\beta = \lambda_{\min}(A)$.

We need one more crucial result (see [28] for a proof), which we state below. This theorem should be of wider interest as it enlarges the class of maps that one can study using the Thompson metric.

**Theorem 11.** *Let $X \in \mathbb{C}^{d \times p}$, where $p \leq d$, and $\text{rank}(X) = p$. Let $A, B \in \mathbb{P}_d$. Then,*

$$\delta_T(X^*AX, X^*BX) \quad \leq \quad \delta_T(A, B). \tag{20}$$

We now show how to use Prop. 10 and Thm. 11 to analyse contractions on $\mathbb{P}_d$.

**Proposition 12.** *Let $h$ be a LN function. Then, the map $\mathcal{G}$ in (17) is nonexpansive in $\delta_T$. Moreover, if $h$ is wlc, then $\mathcal{G}$ is weakly-contractive in $\delta_T$.*

*Proof.* Let $S, R \succ 0$ be arbitrary. Then, we have the following chain of inequalities

$$
\begin{aligned}
\delta_T(\mathcal{G}(S), \mathcal{G}(R)) = \quad & \delta_T\left(\tfrac{2}{n} X h(D_S) X^T, \ \tfrac{2}{n} X h(D_R) X^T\right) \\
\leq \quad & \delta_T\left(h(D_S), h(D_R)\right) \quad \leq \quad \max_{1 \leq i \leq n} \delta_T\left(h(x_i^T S^{-1} x_i), h(x_i^T R^{-1} x_i)\right) \\
\leq \quad & \max_{1 \leq i \leq n} \delta_T\left(x_i^T S^{-1} x_i, x_i^T R^{-1} x_i\right) \quad \leq \quad \delta_T\left(S^{-1}, R^{-1}\right) = \delta_T(S, R),
\end{aligned}
$$

where the first inequality follows from (19b) and Thm. 11; the second inequality follows since $h(D_S)$ and $h(D_S)$ are diagonal; the third follows from (19d); the fourth from another application of Thm. 11; while the final equality is via (19a). This proves nonexpansivity. If in addition $h$ is weakly log-contractive and $S \neq R$, then the second inequality above is strict, that is,

$$\delta_T(\mathcal{G}(S), \mathcal{G}(R)) < \delta_T(S, R) \quad \forall S, R \quad \text{and} \quad S \neq R. \qquad \square$$

Consequently, we obtain the following main convergence theorem for (17).

**Theorem 13.** *If $\mathcal{G}$ is weakly contractive and (16) has a solution, then this solution is unique and iteration (17) converges to it.*

When $h$ is merely LN (not wlc), it is still possible to show uniqueness of (16) up to a constant. Our proof depends on the following new property of $\delta_T$, which again should be of broader interest.

**Theorem 14.** *Let $\mathcal{G}$ be nonexpansive in the $\delta_T$ metric, that is $\delta_T(\mathcal{G}(X), \mathcal{G}(Y)) \leq \delta_T(X, Y)$, and $\mathcal{F}$ be weakly contractive, that is $\delta_T(\mathcal{F}(X), \mathcal{F}(Y)) < \delta_T(X, Y)$, then $\mathcal{G} + \mathcal{F}$ is also weakly contractive.*

Observe that the property proved in Thm. 14 is a striking feature of the nonpositive curvature of $\mathbb{P}_d$; clearly, such a result does not usually hold in Banach spaces. As a consequence, Thm. 14 helps establish the following "robustness" result for iteration (17).

**Theorem 15.** *If $h$ is LN, and $S_1 \neq S_2$ are solutions to the nonlinear equation (16), then iteration (17) converges to a solution, and $S_1 \propto S_2$.*

As an illustrative example of these results, consider the problem of finding the minimum of negative log-likelihood solution of Kotz type distribution. The convergence of the iterative algorithm in (17) can be obtained from Thm. 15. But for the Kotz distribution we can show a stronger result, which helps obtain geometric convergence rates for the fixed-point iteration.

**Lemma 16.** *If $c > 0$ and $-1 < b < 1$, the function $h(x) = x + cx^b$ is weakly log-contractive.*

According to this lemma, $h$ in the iterative algorithm 16 for the Kotz-type distributions with $0 < \beta < 2$ and $\alpha < \frac{d}{2}$ is wlc. Based on Thm. 9, $K(S)$ has a minimum. Therefore, we have the following.

**Corollary 17.** *The iterative algorithm (16) for the Kotz-type distribution with $0 < \beta < 2$ and $\alpha < \frac{d}{2}$ converges to a unique fixed point.*

## 4 Numerical results

We briefly highlight the numerical performance of our fixed-point iteration. The key message here is that our fixed-point iterations solve nonconvex likelihood maximisation problems that involve a complicating hpd constraint. But since the fixed-point iterations always generate hpd iterates, no extra eigenvalue computation is needed, which leads to substantial computational advantages. In contrast, a nonlinear solver must perform constrained optimisation, which can be unduly expensive.

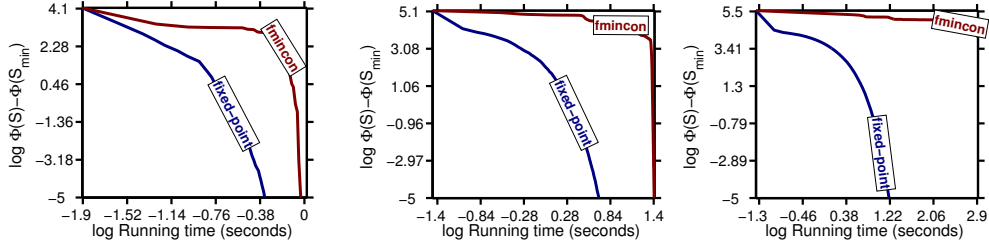

Figure 1: Running times comparison of the fixed-point iteration compared with MATLAB's fmincon to maximise a Kotz-likelihood (see text for details). The plots show (from left to right), running times for estimating $S \in \mathbb{P}_d$, for $d \in \{4, 16, 32\}$. Larger $d$ was not tried because fmincon does not scale.

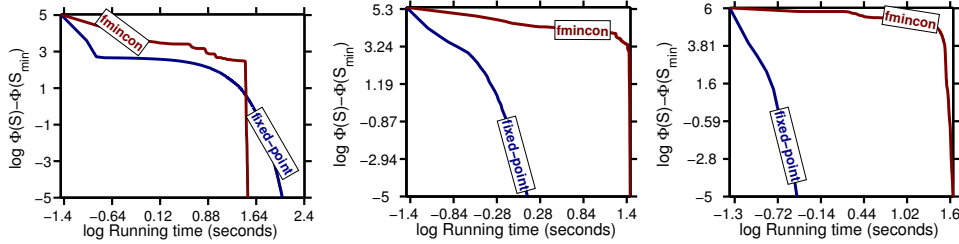

Figure 2: In the Kotz-type distribution, when $\beta$ gets close to zero or 2, the contraction factor becomes smaller which could impact the convergence rate. This figure shows running time variance for Kotz-type distributions with fixed $d = 16$, and $\alpha = 2$, for different values of $\beta$: $\beta = 0.1$, $\beta = 1$, $\beta = 1.7$.

We show two short experiments (Figs. 1 and 2) showing scalability of the fixed-point iteration with increasing dimensionality of the input matrix, and for varying $\beta$ parameter of the Kotz distribution; this parameter influences the convergence rate of the fixed-point iteration. For three different dimensions $d = 4$, $d = 16$, and $d = 32$, we sample 10,000 datapoints from a Kotz-type distribution with $\beta = 0.5$, $\alpha = 2$, and a random covariance matrix. The convergence speed is shown as blue curves in Figure 1. For comparison, the result of constrained optimisation (red curves) using MATLAB'S optimisation toolbox are shown. The fixed-point algorithm clearly outperforms MATLAB'S toolbox, especially as dimensionality increases. These results indicate that the fixed-point approach can be very competitive. Also note that the problems are nonconvex with an open constraint set—this precludes direct application simple approaches such as gradient-projection (since projection requires closed sets; moreover, projection also requires eigenvector decompositions). Additional comparisons in the longer version [28] show that the fixed-point iteration also significantly outperforms sophisticated manifold optimisation techniques [5], especially for increasing data dimensionality.

## 5    Conclusion

We developed geometric optimisation for minimising potentially nonconvex functions over the set of positive definite matrices. We showed key results that help recognise geodesic convexity; we also introduced the class of log-nonexpansive functions that contains functions that need not be g-convex, but can still be optimised efficiently. Key to our ideas here was a careful construction of fixed-point iterations in a suitably chosen metric space. We motivated, developed, and applied our results to the task of maximum likelihood estimation for various elliptically contoured distributions, covering classes and examples substantially beyond what had been known so far in the literature. We believe that the general geometric optimisation techniques that we developed in this paper will prove to be of wider use and interest beyond our motivating application. Developing a more extensive geometric optimisation numerical package is part of our ongoing project.

## Footnotes

[1]To our knowledge the name "geometric optimisation" has not been previously attached to hpd matrix optimisation, perhaps because so far only scattered few examples were known. Our theorems provide a starting point for recognising and constructing numerous problems amenable to geometric optimisation.

[2] For simplicity we describe only mean zero families; the extension to the general case is trivial.

[3]The dgfs of different distributions are brought here for the reader's convenience. Multivariate power exponential: $\phi(t) = \exp(-t^\nu/b), \quad \nu > 0$; Multivariate W-distribution: $\phi(t) = t^{\nu-1}\exp(-t^\nu/b), \quad \nu > 0$; Elliptical gamma: $\phi(t) = t^{\nu-d/2}\exp(-t/b), \quad \nu > 0$; Multivariate t: $\phi(t) = (1 + t/\nu)^{-(\nu+d)/2}, \quad \nu > 0$; Multivariate Pearson type II: $\phi(t) = (1-t)^\nu, \quad \nu > -1, 0 \leq t \leq 1$; Elliptical multivariate logistic: $\phi(t) = \exp(-\sqrt{t})/(1 + exp(-\sqrt{t}))^2$.

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
