[Reviews · NeurIPS 2013]

Submitted by Assigned_Reviewer_6

The bottom line of this paper is an efficient algorithm for finding maximum likelihood estimators for "elliptically contoured distributions", a class of densities that includes the Gaussian and various generalizations of it. For the Gaussian itself, that optimization is straightforward, it's the generalizations where the new algorithm provides real advantages.

One could argue that this focus on a relatively arcane family of distributions (Kotz-type) limits the utility of this paper. But I think it's actually the other way round: The paper may spark new interest at NIPS in these models. Another argument in favour of this paper, entirely separate from this issue, is that it provides several interesting new theorems along the wayside: Two Theorems (1 and 4) helping in the construction and identification of geodesically convex functions, and existence theorems (7 and 9) for the maximum of the likelihood.

I'm not a specialist in this area, so I would be grateful if the authors could answer the following questions to make sure I don't misunderstand:

1) I assume line 278 contains a typo? It should be D_S := Diag[ h(x'_i * S * x_i) ], as opposed to current form of D_S := Diag[ x'_i * S * x_i ] ? With that, since h(A) = I for the Gaussian, iteration 16 for the Gaussian is simply the classic maximum likelihood estimate S = 2/n XX' ? This might be a good mental guardrail to mention at this point in the paper.

2) For the Gaussian, there exists a conjugate prior (the inverse Wishart) for S which turns (4) into a maximum a posteriori estimate and ensure identifiability. Is there a similar solution for the Kotz-type distributions? The reason I'm asking is that this would make Lemma 8 less important, which would be nice since assumption (13) and the requirement that the data span the whole space is restrictive, in particular with regard to nonparametric formulations.
Summary: A theoretical paper about an efficient likelihood maximization method for a rarely studied class of distributions, with several nontrivial theorems as byproducts. Not typical for NIPS, but an interesting read.

Submitted by Assigned_Reviewer_7

The paper develops new and general tools for geodesically convex functions (that is functions defined on C^2 Riemannian manifolds, which are convex "with respect to geodesics", see Definition 3). In particular, the case of the manifold of HD matrices is considered.
Moreover, from the algorithmic point of view, the authors study the fixed point iteration, deriving necessary conditions to ensure that a map is contractive.
In my opinion this is a very good paper.
The paper is very clear and well-written, and organized.
It presents an original point of view to study new and already known results from a more general perspective.

Minor comments:
line 113: it could be interesting to relate geodesic convexity with other notions of generalized convexity (see e.g. invexity)
line 124: I think it i s better to state that P_d is the convex cone of positive definite matrices.
line 287: I think it is not correct to say that (17) is not a fixed point iteration, since it is indeed a fixed point iteration. In my opinion it would be better to say that the map is not contractive with respect to the euclidean metric.
Summary: I think that this paper presents new original ideas which are relevant for optimization over the set of positive definite matrices. Moreover, it is very clearly written.

Submitted by Assigned_Reviewer_8

Summary of the paper
The paper proposes a framework for optimizing (potentially non-convex) functions on the space of hermititian positive definite matrices. This framework makes use of convexity, with respect to geodesics, to empirically estimate the parameters of elliptically contoured distributions based on the minimization of the negative log-likelihood. For this sake, a fixed point algorithm is derived as well as its convergence guarantees. Theoretical tools that help to recognize such geodesical convex functions are given. Finally, numerical experiments illustrate proposed algorithm.


Comments
This paper propose promising results for the optimization of functions subject to Hermitian Positive Definite (HPD) constraints. However the paper has a strong optimization and discrete machine learning flavor and is a bit tedious to read. I would recommend to modify the paper such that the machine learning aspect of this work is pushed forward. Seemingly, the work fits more the topics of a pure optimization conference.

Some nice results about geodesic convexity and log-nonexpansive functions are stated. Those results are used to build a fixed-point algorithm. Even if the derivation seems trivial to the authors, it would be helpful to make explicit the optimization algorithms derived from the theoretical results.

Numerical experiments illustrating the effectiveness of proposed algorithms need to be improved. Indeed, comparison with the matlab function "fmincon" looks impressive, but it would be more interesting to consider state of the art methods such as the gradient (or hessian) based methods developed in the context of optimization over manifolds (see for instance the work "Optimization algorithm on matrix manifolds" by Absil et al.). Comparison to such competitive approaches in terms of convergence speed of the objective function or of the iterates would be appreciable. Besides these experimental evaluations the paper lacks an analysis of the algorithm convergence rate.

Finally, there are some comments about the notations. It is somewhat confusing to refer to HPD matrices both as a cone or as a manifold. Indeed, in those two cases, the implied geometrical structures are different.
Summary: Potentially interesting paper. However it requires some improvements in terms of theoretical presentation, algorithm derivation and empirical comparison.

Submitted by Assigned_Reviewer_9

The paper, 'Geometric optimisation on positive definite matrices for elliptically contoured distributions', addresses the problem of optimizing functions of HPD matrices, with applications to MLEs of elliptically contoured distributions. The authors introduce ECDs and allude to geometric optimization for HPD matrices. Theorem 1 and its corollaries are facts about the matrix geometric mean, while the remainder of section 2 are facts regarding geodesic convexity. Section 3 employs these concepts to ECDs to show that a map (the algorithm proposed) has a fixed point. There are some numerical results showing convergence of the method.

This is a novel, quality paper. The paper is not particularly clear, but it does alert the ML public to these GO methods. If the authors spent more time explaining the significance of the concepts introduced in section 2 then they may have more meaning to the readers.

in the abstract: 'some of our'
either use capitalized acronyms or not (example 338)
fix 412 (10,1000)
Summary: It is a good paper that introduces new concepts, on the other hand, it is difficult to read and it's not clear how the GO concepts will help the ML community solve other problems.
Author Feedback

Author rebuttal: We thank the reviewers for their feedback and suggestions. We address the comments individually below.

As noted by the reviewers, the Geometric Optimisation (GO) framework is the key novelty of the paper as is our new fixed-point theory. Both are then applied to model elliptically contoured distributions (ECDs) in greater generality than was previously possible.

We remark that our ideas combine the convex conic as well as manifold geometry of the HPD matrices in new way: the conic geometry underlies our fixed-point theory, while the manifold view characterises geodesic convexity. We would like to also briefly remark that the widely used class of Geometric Programs (see e.g., the 61pg survey by S. Boyd et al.) is ultimately nothing but the 1D scalar version of GO. The HPD matrix case is, however, harder and requires some more tools because matrices do not commute.

ASSIGNED_REV_6

* "focus ... the other way round"

Thanks for your encouraging words. As noted by you, our theorems that help recognise g-convexity are a key contribution. Beyond g-convexity, our theorems on log-nonexpansive functions show sufficient conditions for efficient fixed-point iterations. Remarkably, some functions may fail to be g-convex yet remain globally optimisable as seen in the paper.

Although ECDs (see also [10,16]) drove us to develop GO, they are not the only application. For example, Thms 11 and 14 extend the scope of the Thompson metric, which allows one to solve a larger variety of problems than previously possible [12].

* "typo on L278"

Thanks! Indeed, this should have an 'h' in it. We agree that mentioning the Gaussian special case here will be useful.

* "conjugate prior...Kotz-type"

We are not aware of derivations of conjugate priors for Kotz-type distributions. We are working to loosen the condition given in (13). We can show that if the data do not satisfy (13), then the fixed point algorithm---for distributions given in Cor. 17---can still converge to a unique fixed point. Although in this case there is an indefinite number of singular matrices for which the likelihood becomes infinite, the fixed-point method converges to a *unique* singular matrix. This procedure can be used for robust subspace recovery.

ASSIGNED_REV_7

Thanks for your encouraging words.

We agree with the suggested 'minor comments' and will incorporate them in the paper; regarding invexity: the key difficulty there is that usually determining whether a given nonconvex function is invex is very hard, short of actually proving that all its stationary points are global minimisers; in contrast, our theorems make recognising g-convexity easier.

ASSIGNED_REV_8

* "not ML ... pure optimization..."

We differ in opinion here as NIPS regularly features cutting edge optimisation research. Moreover, our paper develops novel optimisation techniques that might interest the NIPS audience who regularly deal with HPD kernel and covariance matrices.

However, we do agree that it'll be helpful to add some more examples that highlight ML connections (e.g., to classification, search, vision, kernel methods, etc.; please see the response below)

* "tedious"

The noncommutativity of matrices requires us to use somewhat heavier machinery that may make the paper a bit denser to read.

* "make explicit...algorithms"

This is a nice suggestion---we'll make pseudocode explicit in the paper to aid the reader.

* "fmincon and manifold optimization..."

The primary contribution of our geodesic convexity and fixed-point theorems is to determine when a unique stationary point exists. After existence has been ensured, *any* method---be it a nonlinear solver, gradient / Newton method on the manifold, or our fixed-point iteration---is acceptable as long as it ensures stationarity.

That said, whenever possible, we prefer the fixed-point method due to its simplicity, ease of implementation, and empirical performance. To make the numerical results stronger, we will definitely include additional results in the longer (arXiv) version of our paper.

For example, using the excellent manifold optimisation toolbox from manopt.org on our problem shows that: conjugate-gradient (also trust-regions) on the manifold is faster than fmincon, *but* can easily be 30-50 times slower than our fixed-point iteration.

* "HPD cone and manifold"

The set of HPD matrices is a convex cone, and endowed with a suitable metric, also a Riemannian manifold; our results rely crucially on both geometric views.

ASSIGNED_REV_9

* "more time explaining..."

We will add more discussion to highlight the vast potential GO may have, especially as it includes Geometric Programming as a 1D special case.

We mention below a few additional works on classification, clustering, computer vision, and statistics; these rely implicitly or explicitly on GO concepts.

[a] Harandi et al; ECCV 2012. Sparse coding and dictionary learning for symmetric positive definite matrices: A kernel approach

[b] Cherian et al. IEEE TPAMI 2012. Jensen-Bregman LogDet Divergence for Efficient Similarity Computations on Positive Definite Tensors

[c] Chen et al. IEEE TSP 2011. Robust shrinkage estimation of high-dimensional covariance matrices.

Beyond these, the survey paper [16], and refs [20,21] contain other relevant applications.